# The Neurotrophic Function of Glucagon-Like Peptide-1 Promotes Human Neuroblastoma Differentiation via the PI3K-AKT Axis

**DOI:** 10.3390/biology9110348

**Published:** 2020-10-22

**Authors:** Jenq-Lin Yang, Yu-Ting Lin, Wei-Yu Chen, Yun-Ru Yang, Shu-Fang Sun, Shang-Der Chen

**Affiliations:** 1Institute for Translation Research in Biomedicine, Kaohsiung Chang Gung Memorial Hospital, 123 Ta Pei Road, Kaohsiung City 83301, Taiwan; jyang@cgmh.org.tw (J.-L.Y.); wychen624@cgmh.org.tw (W.-Y.C.); s30705ruby@gmail.com (Y.-R.Y.); s981537101@cgmh.org.tw (S.-F.S.); 2Department of Pharmacology, College of Medicine, National Cheng-Kung University, Tainan 70101, Taiwan; t74mp@yahoo.com.tw; 3Department of Neurology, Kaohsiung Chang Gung Memorial Hospital, 123 Dapi Road, Kaohsiung City 83301, Taiwan

**Keywords:** GLP-1, PI3K-AKT axis, neuroblastoma differentiation

## Abstract

**Simple Summary:**

The study demonstrated that the treatment with GLP-1 of SH-SY5Y human neuroblastoma cells increased the expression of AMPA receptors, NMDA receptors, dopamine receptors, synaptic proteins-synapsin 1, synaptophysin, and postsynaptic density protein 95, but not muscular and nicotinic acetylcholine receptors. In addition, the biomarker of dividing neuronal cells, vimentin, was decreased after treatment with GLP-1. Tuj1 immunostaining images showed that GLP-1 induced neurite processes and the development of neuronal morphologies. The GLP-1-differentiated neurons were able to be induced to generate action potentials by single cell patch-clamp. Our results also suggested that the PI3K-AKT axis is the dominant signaling pathway promoting the differentiation of SH-SY5Y cells into mature and functional neurons in response to GLP-1 receptor activation. The sequential treatment of retinoic acid and GLP-1 within a serum-free medium is able to trigger the differentiation of SH-SY5Y cells into morphologically and physiologically mature glutamatergic and dopaminergic neurons.

**Abstract:**

***Background:*** Neurons are terminally-differentiated cells that generally develop from neuronal stem cells stimulated by various neurotrophic factors such as NGF, BDNF, NT3, and NT-4. Neurotrophic factors have multiple functions for neurons, including enabling neuronal development, growth, and protection. Glucagon-like peptide-1 (GLP-1) is an intestinal-secreted incretin that enhances cellular glucose up-take to decrease blood sugar levels. However, many studies suggest that the function of GLP-1 is not limited to the regulation of blood sugar levels. Instead, it may also act as a neurotrophic factor with a role in ensuring neuronal survival and neurite outgrowth, as well as protecting synaptic plasticity and memory formation. ***Methods:*** The SH-SY5Y cells were differentiated by sequential treatments of retinoic acid and GLP-1 treatment within polyethylenimine-coated dishes under serum-free Neurobasal medium. PI3K inhibitor (LY294002) and MEK inhibitor (U0126) were used to determine the signaling pathway in regulation of neuronal differentiation. Neuronal marker (TUJ1) and synaptic markers (synapsin 1, synaptophysin, and PSD95) as well as single cell patch-clamp were applied to determine maturity of neurons. Antibodies of AMPA receptor, NMDA receptor subunit 2A, dopamine receptor D1, muscarinic acetylcholine receptor 2, and nicotinic acetylcholine receptor α4 were used to examine the types of differentiated neurons. ***Results:*** Our study’s results demonstrated that the treatment with GLP-1 of SH-SY5Y human neuroblastoma cells increased the expression of AMPA receptors, NMDA receptors, dopamine receptors, synaptic proteins-synapsin 1, synaptophysin, and postsynaptic density protein 95, but not muscular and nicotinic acetylcholine receptors. In addition, the biomarker of dividing neuronal cells, vimentin, was decreased after treatment with GLP-1. Tuj1 immunostaining images showed that GLP-1 induced neurite processes and the development of neuronal morphologies. The GLP-1-differentiated neurons were able to be induced to generate action potentials by single cell patch-clamp. Our study also suggested that the PI3K-AKT axis is the dominant signaling pathway promoting the differentiation of SH-SY5Y cells into mature and functional neurons in response to GLP-1 receptor activation. ***Conclusions:*** The sequential treatment of retinoic acid and GLP-1 within a serum-free medium is able to trigger the differentiation of SH-SY5Y cells into morphologically and physiologically mature glutamatergic and dopaminergic neurons.

## 1. Introduction

### 1.1. Glucagon-Like Peptide 1 (GLP-1)

Glucagon-like peptide-1 (GLP-1), an incretin hormone secreted from enteroendocrine L-cells of the small intestine, was first identified in the early 1980s [1]. Incretins belong to a group of metabolic hormones that are secreted from the intestine and decrease blood glucose levels in the body. Incretin hormones with a blood glucose-dependent mechanism are released with food intake and increase the secretion of insulin from pancreatic beta cells, which also slows the gastric emptying time, reduces food intake, and decreases the absorption rate of nutrients into the blood stream. Furthermore, incretins are able to inhibit glucagon release from pancreatic alpha cells [2]. Within minutes of food intake, the active forms of GLP-1 are released into the circulatory system for the body’s needs and are rapidly degraded after secretion by the enzyme dipeptidyl peptidase-4 (DPP-4) [2,3,4]. GLP-1 is also suggested to have trophic properties. For instance, it triggers islet β cell differentiation and proliferation, inhibits apoptosis, and enhances cell survival [5,6]. Clinically, DPP-4 inhibitors, a class of oral hypoglycemics that block DPP-4 activity, increase GLP-1 concentrations and are used to treat type 2 diabetes mellitus [7]. The actions of GLP-1 are mediated by the GLP-1 receptor (GLP-1R), a member of the glucagon receptor family with a transmembrane spanning GTPase protein [8]. Currently, many agonists of GLP-1 receptors have been developed and applied for the clinical treatment of type 2 diabetes [9,10]. Except for its effects in controlling hyperglycemia in diabetic patients, GLP-1R agonists have been suggested as potential neuroprotective agents for neurologic disorders such as Alzheimer’s disease, Parkinson’s disease, cerebral ischemia, and traumatic injury in preclinical and clinical studies [11,12,13,14]. GLP-1R is not only distributed in various peripheral tissues but is also widely expressed in many areas of the CNS [15,16]. The mechanisms by which GLP-1R agonists are neuroprotective are still not well understood, but may be linked to several processes, such as anti-oxidation, anti-inflammation, enhancing DNA repair ability, increasing mitochondrial function and genesis, augmenting synaptic plasticity, decreasing protein aggregation, promoting neurite growth, and reducing apoptosis [17,18,19,20]. In contrast, a study by During et al. reported that GLP-1R knock-out mice have been shown to have defective abilities in learning and memory as alongside being vulnerable to kainite-induced neuronal toxicity [21]. The stimulation of GLP-1Rs has been well known to activate two major signaling pathways: the phosphatidylinositol 3 kinase (PI3K)-protein kinase B (AKT) pathway and the adenosine cyclase (AC)-protein kinase A (PKA)-mitogen-activated protein kinase kinase (MEK)-extracellular-signal-regulated kinase (ERK/MAPK) pathway [22,23]. Another signaling pathway is the glycogen synthase kinase 3β (GSK-3β) pathway [11], which is also reportedly involved in GLP-1R activation. All downstream signaling pathways lead to the phosphorylation of cyclic AMP response element binding protein (CREB) for further transcriptional regulation.

Emerging evidence has revealed that GLP-1R agonists possess trophic characteristics. GLP-1R agonists can trigger islet β cell differentiation and proliferation and improve cell survival [5,6,24]. A similar trophic function is also observed and validated in neuronal tissues. Like insulin, GLP-1 has direct trophic effects on the nervous system. Exendin-4, a GLP-1 analogue, induced neurite outgrowth and cell survival in a manner similar to nerve growth factor (NGF) in neuron-like pheochromocytoma (PC12) cells, sympathetic neurons, and adult sensory neurons in vitro [25,26]. A recent study showed that Exendin-4 raises the expression levels of the NGF gene in the hippocampus of a diabetic mouse model [27]. The study of Spielman in 2017 also demonstrated that microglia express functional GLP-1 receptors, and that GLP-1 increases brain derived neurotrophic factor, glial cell-line derived neurotrophic factor, and nerve growth factor in a PI3K-PKA-dependent manner [28]. A recent review article from Müller et al. provides a thorough overview on the multifaceted roles of GLP-1, which covers classical functions to emerging roles of GLP-1 including gastrointestinal hormones, glucose metabolism, cardiovascular and cardiometabolic effects, regulation of food-intake and body-weight, energy expenditure, regulation of the hypothalamic pituitary-adrenal axis, effects on learning, memory, and neuroprotection, effects on bone, as well as pharmarcological applications [29].

### 1.2. Human Neuroblastoma SH-SY5Y Cells

SH-SY5Y is a derived cell line from SK-N-SH cells that was established from a metastatic bone marrow neuroblastoma in a four-year-old girl in 1970 [30]. The SK-N-SH cells have two morphologically distinct phenotypes: neuroblast-like and epithelial-like cells. The SH-SY5Y cell-line was derived from a neuroblastic subclone of the SK-N-SH line [31]. SH-SY5Y cells have been popularly used for in vitro studies in the field of neuroscience since the cell line was first revealed. An important characteristic of SH-SY5Y cells is their ability to be differentiated into more mature neuron-like phenotypes by manipulating ingredients of the culture medium, thereby facilitating research studying cellular and molecular mechanisms in neurons. Several advantages of SH-SY5Y cells in neuroscience research have generally included their relatively easy, fast, cheap, and large-scale expansion to produce differentiated neurons, as compared to stem cell differentiated and primary cultured neurons [31]. In addition, SH-SY5Y cells are human-derived, enabling the differentiated neurons to be studied in the context of human neuronal disorders without incurring any ethical issues. In addition, they have conveniently synchronized cell-cycles in their differentiation into homogenous neuronal cells [32,33]. Therefore, developing a standard procedure to differentiate mature and functional neurons from mitotic neuronal cell lines and to characterize the differentiated neurons are important issues enabling the facilitation and acceleration of neuroscience research.

In this study, we sought to investigate the potential neurotrophic effect of GLP-1 in neuronal differentiation and the critical signaling pathway involved. We also assessed various neuronal markers to better understand what types of neurons GLP-1-tends to induce differentiation in. Our results have shown that the stimulation of GLP-1 receptors results in the differentiation of SH-SY5Y cells into morphologically mature and physiologically functional neurons. Moreover, GLP-1-differentiated neurons tend to be glutamatergic and dopaminergic neurons.

## 2. Materials and Methods

### 2.1. Cell Culture and Treatments

SH-SY5Y cells are human neuroblasts derived from a bone marrow neuroblastoma. They were purchased from the American Type Culture Collection (ATCC). SH-SY5Y cells were maintained in high glucose Dulbecco’s Modified Eagle Medium (DMEM, Gibco^TM^, Waltham, MA, USA) with 10% FBS (GibcoTM) and 1% Penicillin-Streptomycin (Gibco^TM^, Waltham, MA, USA) under conditions of 5% CO_2_ and 37 °C. SH-SY5Y cells were cultured with 10 µM of retinoic acid (Sigma-Aldrich, St. Louis, MO, USA) and maintained in medium for 5 days to initiate the differentiation process. Following the retinoic acid (RA) treatment, cells were seeded on the 0.01% polyethylenimine-coated petri dishes with RA in the maintenance medium. After the cells attached, the medium of subcultured SH-SY5Y cells was replaced with Neurobasal (Gibco^TM^, Waltham, MA, USA) with B27 supplement (GibcoTM), 4 mM glutamine, 0.01% gentamycin sulfate (GibcoTM), and 100 nM GLP-1 (Sigma-Aldrich) being used for assessing neuronal differentiation. To determine how GLP-1 downstream signaling affects SH-SY5Y neuronal differentiation, a 10 μM MEK inhibitor (U0126, Cell Signaling, Danvers, MA, USA) or a 20 μM PI3K inhibitor (LY294002, Cell Signaling, Danvers, MA, USA) was added into GLP-1 containing Neurobasal medium while the RA contained DMEM was replaced. The cells were harvested or fixed on 5-day RA treatment, day 1, day 3, day 5, and day 7 post-inhibitor treatment for further analyses of immunoblotting or immunofluorescent labelling.

### 2.2. Reverse Transcription-Polymerase Chain Reaction (RT-PCR)

The detection of GLP-1R mRNA of SH-SY5Y cells was carried out using the forward primer: 5′-TCC TTC CAG GTG ACT TCA TGC-3′; reverse primer: 5′-TGG TAC CCA GAG CTA CAT CCA-3′ for RT-PCR. The GAPDH mRNA was used as a house keeping gene control, using the forward primer: 5′-ACC CCT TCA TTG ACC TCA AC-3′ and the reverse primer: 5′-TCG CTC CTG GAA GAT GGT GAT-3′. The total RNA extraction of SH-SY5Y cells and cDNA generation were carried out using the one-step “Direct RT-PCR” kit from Yeastern Biotech Co., Ltd. (Taipei, Taiwan).

### 2.3. Immunoblotting

Cultured cells were extracted in RIPA buffer (150 mM NaCl, 0.1% SDS, 0.5% sodium deoxycholate, 1X protease inhibitor cocktail (Roche), phosphatase inhibitor cocktail (Pierce), and 50 mM Tris; pH 8.0). Total protein within the lysate was determined according to the BCA™ protein assay kit (Thermo Scientific, Waltham, MA, USA). The detection and dilution factors of the primary antibodies were as follows: vimentin (1:10,000; Abcam, Cambridge, UK), nicotinic acetylcholine receptor α4 (1:10,000; Abcam), muscarinic acetylcholine receptor 2 (1:1000; Abcam), AMPA receptor (1:250; Abcam), NMDA receptor subunit 2A/NMDAR2A (1:1000; Cell Signaling), dopamine receptor subunit D1 (1:5000; Abcam), synapsin 1 (1:2000; Cell Signaling), synaptophysin (1:10,000; Proteintech, Rosemont, IL, USA), postsynaptic density protein 95/PSD95 (1:1500; BD Bioscience, Heidelberg, Germany), acetylcholinesterase/AChE (1:500; Santa Cruz, Dallas, TX, USA), and actin (1:5000; Sigma-Aldrich). The horseradish peroxidase-conjugated secondary antibodies (Invitrogen) were diluted to a ratio of 1:2000–10,000. Immunolabeled proteins were visualized by using an enhanced chemiluminescence kit (Amersham). Images were taken and analyzed by the UVP ChemStudio PLUS Touch system (Analytik Jena).

### 2.4. Immunofluorescent Labeling

Cultured SH-SY5Y cells or differentiated SH-SY5Y neurons were briefly washed in cold 1XPBS and fixed immediately with 4% paraformaldehyde in phosphate buffer saline (PBS) for 10 min. The formaldehyde fixed cells were incubated with a permeabilization buffer (0.5% Triton X-100, 100 mM glycine, 1% BSA, 0.7 mM EDTA) for 10 min on ice. The samples were then incubated with a blocking buffer (10% bovine serum, 0.01% sodium azide in 1X PBS) at 37 °C for 1 h or 4 °C overnight. Diluted primary antibodies (TUJ1, Sigma-Aldrich, 1:500; GLP-1R, Santa Cruz, 1:500) were incubated with samples for 1 h at 37 °C. The samples were then rinsed with wash buffer (0.05% Tween-20 in 1x PBS) three times for 5 min at room temperature. A fluorophore conjugated secondary antibody (Alexa series, Invitrogen, Carlsbad, MA, USA; 1:1000 dilution) was added and the samples were incubated for 1 h at 37 °C. Samples were rinsed briefly with wash buffer (3 × 5 min) and mounted with Prolong Gold containing DAPI (nuclear marker, Invitrogen). Images of immunostained neurons were captured using an Olympus FV10i confocal microscope system. The ImageJ (NIH) was used to quantitate the length of differentiated neurites.

### 2.5. Electrophysiological Assay

Whole cell patch-clamp recordings were carried out to determine the electric activity of GLP-1 treated SH-SY5Y cells. Coverslips with differentiated SH-SY5Y cells were constantly perfused with an external solution (140 mM NaCl, 3 mM KCl, 2 mM CaCl_2_, 1.3 mM MgCl_2_, 10 mM HEPES, and 10 mM glucose, pH 7.4) at a rate of 2–3 mL/min. Recording micro-pipettes were filled with an internal solution (120 mM K-gluconate, 15 mM KCl, 10 mM HEPES, 4 mM MgCl_2_, 0.1 mM EGTA, 4 mM Na_2_ATP, 0.3 mM Na_3_GTP, 7 mM phosphocreatine, pH 7.3). To examine neuronal-like properties, current pulses of 500 ms in 0.2 nA steps were delivered in current-clamp mode and the peak of the first fired action potential was measured. Digidata 1440A digitizer (Molecular Devices) and pCLAMP 9 software (Molecular Devices) were used for data acquisition and analysis.

### 2.6. Measurement of Neuronal Viability

Two methods were used to determine cell viability. The Premix WST-1 Cell Proliferation Assay System, purchased from Takara Bio Inc. (Shiga, Japan), was used to determine cell viability for various treatments. The assay procedure was carried out according to the protocol in the user’s manual. We also directly counted neurons on the same spotted field using the Olympus CellR fluorescent microscopy system. The 5-day RA-preconditioned SY-SY5Y cells were plated and treated with GLP-1 and inhibitors in polyethylenimine-coated 4-well chamber slides as described above. Images of differentiating SH-SY5Y cells in designated microscope fields (10× objective) were acquired post-GLP-1 treatment on day 0, day 1, day 3, day 5, and day 7. The cell survival after the inhibitor treatment at each time point was expressed as a fold of the coordinating time of GLP-1 treated cells.

### 2.7. Statistics

Statistical comparisons were made using a Student’s *t*-test. A one-way ANOVA was used to evaluate the significance of the differences between groups, after which a Bonferroni *t*-test was used for multiple comparisons (* *p* < 0.05, ** *p* < 0.01, *** *p* < 0.001). All values in graphs represent the mean and standard error (SE) of determinations from at least 4–6 separate experiments.

## 3. Results

### 3.1. Cortical Neurons Express GLP-1 Receptor (GLP-1R) and GLP-1 Triggers Neuronal Differentiation

Reverse-transcription PCR (RT-PCR) and Western blotting techniques were used to detect the presence of GLP-1 receptor (GLP-1R) in SH-SY5Y human neuroblastoma cells. Both mRNA and protein levels of GLP-1R were detected. The total RNA from SH-SY5Y cells was harvested and one-step RT-PCR was used to assess GLP-1R mRNA (Figure 1b). The results of Western blotting demonstrated that GLP-1R proteins were also expressed in cortical neurons (Figure 1c). Furthermore, we used immunocytochemistry against GLP-1R and neuron-specific class III beta-tubulin (TUJ1) to assess the presence and distribution of GLP-1R and the status of neuronal expression in SH-SY5Y cells. Fluorescent microscopy images have shown that GLP-1R was mostly located in the cell membrane and that TUJ1 was only slightly expressed in the cytoplasm (Figure 1a). The images suggested that original SH-SY5Y cells express GLP-1 receptors without any mature neuronal traits. Furthermore, to test whether the GLP-1 better induces the differentiation of SH-SY5Y cells than a previous study [34], we carried out an additional 5-day retinoic acid (RA) precondition and replaced the culture medium to serum-free Neurobasal with B27 supplement. The TUJ1 immunofluorescent staining images were recorded and quantitated during the GLP-1 treatment period, demonstrating neurite outgrowth according to the GLP-1 treatment durations (Figure 2). The results indicated that RA-preconditioning and GLP-1 administration time-efficiently differentiated SH-SY5Y cells into morphologically mature neurons (Full western blot please check Appendix A).

### 3.2. GLP-1 Receptor Stimulation Inhibits Vimentin Expression and Promotes the Expression of AMPA, NMDA, and DA Receptors in SH-SY5Y Cells

Based on previous results, we sought to assess vimentin, a type III intermediate filament protein expressed in many types of cells. Vimentin is characterized by its lack of specificity, which is a result of its wide distribution in a variety of cell types in early embryonic life, before these are replaced by more specific intermediate filaments later in development. Therefore, vimentin is commonly seen in neuroblastoma and neural progenitor cells and was used to assess neuronal maturity as reflected in its reduced presence. A five-day RA-treatment upregulated vimentin expression in SH-SY5Y cells suggested that neuroblastoma cells start differentiating into neural progenitor cells. Following the 24-h GLP-1 treatment, the RA-preconditioned SH-SY5Y cells turned into neural progenitor-like cells that express high levels of vimentin, but continuing GLP-1 treatment significantly downregulated vimentin expression, which almost vanished by the 7th day of treatment (Figure 3a). The vanished expression of vimentin means that GLP-1 treated SH-SY5Y cells had a less mitotic and more differentiated/mature neuron-like status. In the meantime, various neuronal receptors were also assessed by immunoblotting, including α-amino-3-hydroxy-5-methyl-4-isoxazolepropionic acid (AMPA) receptors, N-methyl-D-aspartate (NMDA) receptors, dopamine receptors, muscarinic acetylcholine receptors, and nicotinic acetylcholine receptors. The results of the immunoblotting assay indicated that AMPA, NMDA, and dopamine receptors were all robustly upregulated by GLP-1 treatment (Figure 3b,c). Interestingly, both the muscarinic acetylcholine receptors and nicotinic acetylcholine receptors were downregulated after the 7-day GLP-1 treatment (Figure 3d,e). The results of the immunoblotting assay suggested that GLP-1 induced SH-SY5Y neuronal differentiation into glutamatergic and dopaminergic neurons, but not acetylcholinergic neurons.

### 3.3. GLP-1 Upregulates Synaptic Proteins and Promotes the Differentiation to Electrophysiologically Functional Neurons

Synapses are essential to enabling signaling transmission to another neuron or target cells, and are only found in mature neurons. Hence, the synaptic protein synapsin 1, synaptophysin, and PSD95 were utilized to determine the maturation of GLP-1 treated SH-SY5Y cells. Synaptophysin is a presynaptic vesicle glycoprotein participating in transmission at synaptic vesicles which is uniquely found in the synapses of all neurons. As such, it is commonly used to quantify synapses by Western blotting and immunostaining. Synapsin 1 is a phosphoprotein belonging to the synapsin family which is found at the membranes of synaptic vesicles in the axon terminal. Synapsin 1 can be phosphorylated by Ca^2+^/calmodulin-dependent protein kinases, and phosphorylated synapsin 1 is dissociated from synaptic vesicles to regulate the release of neurotransmitters in the context of neurotransmission. Both synapsin 1 and synaptophysin are used to measure the formation of axon terminals and maturation of neurons. The other synaptic protein, postsynaptic density protein 95 (PSD95), is a member of the membrane-associated guanylate kinase family, which is almost exclusively located in the post synaptic density of neurons. PSD95 is involved in anchoring synaptic proteins [35] and directly or indirectly binding with NMDA receptors, AMPA receptors, or potassium channels in the dendritic terminals of neurons [36]. PSD95 is also used to assess the formation of dendritic terminals and maturation of neurons. The results of the immunoblotting assay revealed that three synaptic proteins were upregulated starting following the 5-day RA conditioning, and that continuous GLP-1 administration kept increasing the expression levels of synapsin 1 (Figure 4a), synaptophysin (Figure 4b), and PSD95 (Figure 4c).

Except for assessing the development of neuronal receptors and synapses in response to GLP-1 treatment, the action potential is another indicator of neuronal function and maturity. The patch-clam technique is a useful electrophysiological tool to examine ion channel behavior as well as neuronal physiological function and maturation. Ten cells of each RA-conditioned, 3-day GLP-1 post-treated, and 7-day GLP-1 post-treated SH-SY5Y group were examined by single cell current-clamp to assess the responsiveness of the action potential (AP) in light of the neuron’s maturation level. The 5-day RA-treated only cells showed very little membrane potential change in response to current-stimuli, the 3-day GLP-1 post-treated cells showed partial membrane potential changes in response to the first stimulus but little or no response to the second stimulus, and the 7-day GLP-1 post-treated cells showed a complete action potential pattern in response to stimuli (Figure 5a). In addition, the single cell patch-clamp technique was also applied to determine the neuronal maturation of GLP-1 administrated cells. The undifferentiated cells (5-day RA-treatment) showed no stimulus response (upper panel of Figure 5b), but differentiated cells (RA-conditioned with continuing GLP-1 treatment) responded to single or multiple stimuli (lower panel of Figure 5b). The differentiated cells of various GLP-1 treatment periods were randomly picked to assess maturation and action potentials. Patch-clamp stimulation showed 54.55%, 76.74%, 85.71%, and 86.05% cells having action potentials on days 5, 7, 10, and 14 of the GLP-1 treatment period, respectively (Figure 5c). The results revealed that longer GLP-1 treated cells obtained better differentiated/mature neurons. In the meantime, we found that approximately half of the mature neurons had multiple-action-potential responses from 7-day to 14-day GLP-1 treatments, but 5-day treatments resulted in only 20% of the mature cells having multiple-action-potential responses (Figure 5c). The multiple-action-potential responses in differentiated neurons suggest that these are more mature and have more active receptors/ion channels.

### 3.4. GLP-1R Activation Induces the PI3K-AKT Signaling Axis and Regulates the Neuronal Differentiation of Human Neuroblastoma Cells

Two major downstream signaling pathways associated with a neuroprotective effect, AC-PKA-MEK-ERK and PI3K-AKT, are triggered after the stimulation of GLP-1R in neurons [20]. We postulated that these two signaling pathways also mediated the neuronal differentiation of SH-SY5Y cells. To test this hypothesis, the specific inhibitors of MEK and PI3K, U0126, and LY294002, were applied to determine which is the dominant signaling axis regulating neuronal differentiation. Immunocytochemistry images revealed that administration of a PI3K inhibitor, LY294002, obviously compromised the GLP-1 differentiating effect, prohibited cell growth, and caused significant cell death (Figure 6a,c). However, the MEK inhibitor, U0126, did not affect GLP-1 induced neuronal differentiation, but still caused a certain amount of cell death (Figure 6b,c). The results of the immunocytochemistry and cell viability assays suggested that the PI3K-AKT-CREB axis is the dominant signaling pathway regulating neuronal differentiation in response to GLP-1 administration.

## 4. Discussion

Retinoic acid (RA) is one of the earliest chemicals applied to induce the neuronal differentiation of SH-SY5Y cells [37]. Later studies have shown that SH-SY5Y cells continuously treated with RA and NGF or BDNF differentiate into and maintain a better mature neuronal phenotype [32,38]. Some other studies have reported that cultured SH-SY5Y cells with a Neurobasal medium with B27 supplement underwent a better differentiation process [31,39]. Earlier studies have demonstrated that agonists of GLP-1Rs induce neurite outgrowth of pheochromocytoma (PC12) cells [26,40] and human neuroblastoma cells (SH-SY5Y) [32,39]. Each study has grown and differentiated cells in different medium conditions, but all studies differentiated cells in low serum medium, such as 2% FBS [34,41], 0.5% FBS [26], or 1% FBS and 2% horse serum [40]. This study consolidated previous cell-culture protocols, ensuring that the 5-day RA-conditioned SH-SY5Y cells were subcultured in polyethylenimine (PEI) coated dishes with GLP-1 contained Neurobasal and B27 supplement. RA seems only to promote partial differentiation, and does not suppress proliferation. In combination with GLP-1, neuronal differentiation reaches completion more readily and cell proliferation is significantly suppressed. Neurite processes and neuronal morphologies develop better in serum-free Neurobasal and B27 medium with PEI coated dishes than per previous protocols. Our immunostaining results showed that the neurotrophic effect of GLP-1 and modified differentiation protocol induced SH-SY5Y cells to develop long and firm neurites as well as electrophysiologically functional neurons in modified culture conditions (Figure 2).

GLP-1R stimulation-induced neuronal differentiation may function via various signaling axes. The study of Perry et al. reported that the downstream signaling pathways of GLP-1R, PI3K-AKT, and MEK-ERK involve neurite outgrowth in GLP-1- and EX-4-treated PC12 cells [26]. Another similar study from Liu et al. also found that a novel agonist of GLP-1R, geniposide, induces neuronal differentiation in PC12 cells via the MEK/MAPK-ERK signaling pathway [40]. In addition, a study of Luciani et al. reported that the specific inhibitor of PI3K, LY294002, completely abolished the neuronal differentiation of SH-SY5Y cells [34]. Our study also demonstrated that PI3K-AKT is a critical signaling pathway regulating GLP-1-induced SH-SY5Y differentiation (Figure 7). In addition, the specific inhibitor of PI3K, LY294002, co-treated with GLP-1, not only suppressing neurite-outgrowth, but also led to significant cell death (Figure 6). The combined treatment with RA and exendin-4 did not have any additive effects in Luciani’s study [34], but our immunostaining results showed that a sequential RA and GLP-1 treatment significantly induced more neurite-outgrowth than the RA-only treatment (Figure 2). The difference between the two studies lies in the differentiation of SH-SY5Y cells under low-serum (2% FBS) or serum-free conditions, suggesting that serum may exert a suppressing function during in vitro neuronal differentiation. All lines of evidence suggest that PI3K-AKT signaling axis is crucial in GLP-1-induced neuronal differentiation; moreover, it implies that enhancing PI3K-AKT also is a potential treatment for neurodegenerative diseases.

In this study, we found that GLP-1-induced SH-SY5Y neurons had increased expression levels of AMPA, NMDA, and DA receptors, but that both nicotinic and muscarinic acetylcholine receptors were decreased compared with the control group (Figure 3). These results suggested that GLP-1-treated SH-SY5Y cells tended to differentiate into glutamatergic and dopaminergic neurons. RA is the most common differentiating agent in SH-SY5Y cells, resulting in partial proliferation inhibition and the enhanced expression of noradrenaline, acetylcholinesterase, choline acetyl transferase, vesicular monoamine transporter, muscarinic receptors, and dopaminergic receptors [32,37,42,43,44]. Evidence has demonstrated that an RA treatment induces SH-SY5Y cells to differentiation into adrenergic, cholinergic, and dopaminergic neurons. Incorporating our results with those of previous studies, the sequential treatment of RA and GLP-1 preferably induces differentiation to glutamatergic and dopaminergic neurons, abolishing cholinergic neurons from RA-induced differentiation but still maintaining dopaminergic neurons (Figure 7). The study of Encinas and colleagues showed that the sequential treatment of RA and BDNF significantly decreases noradrenaline expression compared with RA-treated and naive SH-SY5Y cells [32]. The results of these studies suggest that SH-SY5Y cells could be differentiated into different types of neurons by various treatment conditions or neurotrophic factors. More interestingly, the chronic stimulation of GLP-1Rs by a long-lasting agonist, liraglutide, significantly increases cell proliferation and survival while increasing neuroblast differentiation in both wild type and AD mice [45]. Neurogenesis has been studied for many decades and is thought to be a therapeutic target for neurodegenerative diseases, brain stroke, and brain injuries. Nevertheless, the process of promoting neuronal differentiation from neuronal progenitor cells has not yet been fully understood. This in vitro study has demonstrated that the stimulation of GLP-1Rs has the potential to induce neuronal differentiation into mature glutamatergic and dopaminergic neurons, and may also represent a potential treatment for AD and PD. However, application of GLP-1 or its analogues to induce differentiation of glutamatergic and dopaminergic neurons from in vitro to in vivo for treating AD and PD is still an underlying gap. The limitations include how efficient GLP-1 analogues penetrate the blood–brain barrier; how GLP-1 analogues are effectively delivered to the brain; and what concertation of drugs is used to avoid coacting or counteracting with other organs and tissues, etc. More studies need be pursued to find out answers for the clinical treatment in neurodegenerative diseases.

## 5. Conclusions

Using mitotic cell line differentiated neurons instead of primary neurons for neuroscience studies saves not only on money and time, but also circumvents ethical issues and the difficulty inherent to sampling experiments. In the study, we developed a new differentiation procedure (see Cell Culture and Treatments of Methods section) consolidating from previous studies, which provide a better methodology mimicking mature and functional neurons to facilitate better neuroscience studies. Our results also suggest that the PI3K-AKT signaling axis is crucial in GLP-1-induced neuronal differentiation; moreover, the SH-SY5Y cells prefer to differentiate into glutamatergic and dopaminergic neurons under our culture condition. These results imply that enhancing PI3K-AKT signaling of neuronal progenitor cells is a potential treatment for AD and PD.

## Figures and Tables

**Figure 1 biology-09-00348-f001:**
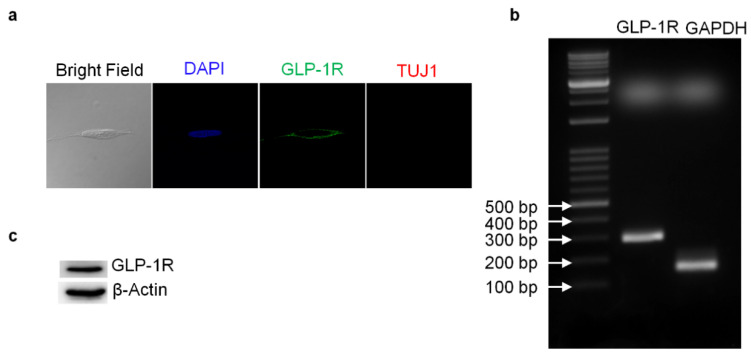
GLP-1 receptor expression detected in the human neuroblastoma SH-SY5Y cells. The confocal images of immunofluorescent staining revealed that SH-SY5Y cells express GLP-1 receptors (GLP-1Rs) which are mainly distributed in cell membrane areas. In addition, the neuron marker Tuj1 (neuron-specific class III beta-tubulin) was slightly expressed in neuroblastoma cells (**a**). The agarose gel electrophoresis image reflecting the reverse transcription-PCR results demonstrated that the mRNA of GLP-1R was expressed in SH-SY5Y cells (**b**). The Western blotting results confirmed that GLP-1Rs were expressed in human neuroblastoma SH-SY5Y cells as well (**c**).

**Figure 2 biology-09-00348-f002:**
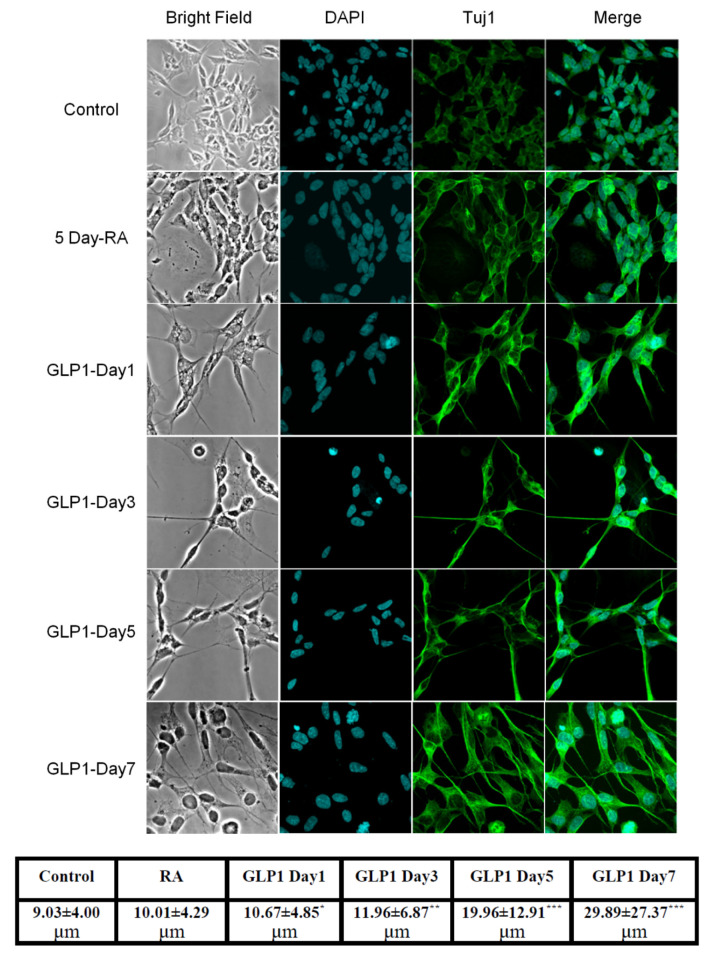
GLP-1 promoted SH-SY5Y cell differentiation into morphologically mature neurons. TUJ1 immunocytochemistry images demonstrated that only 5-day retinoic acid (RA) treated SH-SY5Y cells showed differentiating short neurites (second row); RA-preconditioned SH-SY5Y cells grew longer neurites and altered spindles, developing round cell bodies, after GLP-1 treatment. According to the treatment time with GLP-1, neuroblastoma had differentiated into mature neuron-like morphologies. The bottom table demonstrates quantitated average length of neurites in each group. The GLP-1 treated SH-SY5Y cells grew longer neurites on the 3rd day, and almost all cells had significantly longer neurites on the 7th day post-treatment. (M ± SD, the statistical significance was compared with value of control group, * *p* < 0.05; ** *p* < 0.01; *** *p* < 0.001, *n* = 50).

**Figure 3 biology-09-00348-f003:**
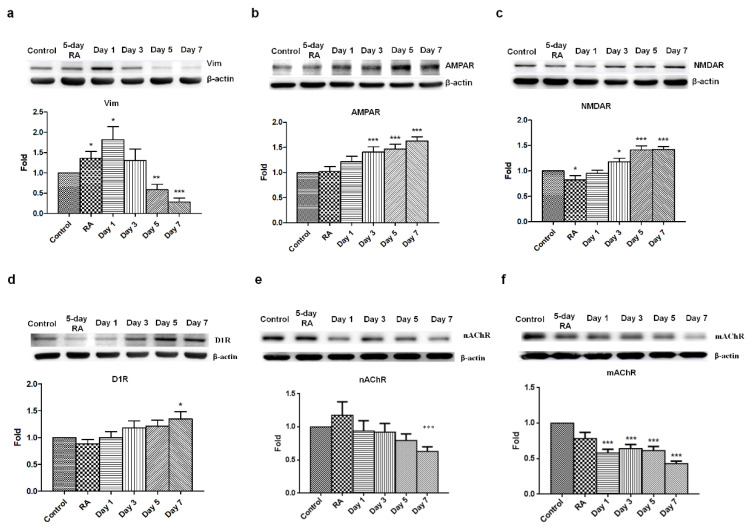
Decreased vimentin expression and promoted AMPA, NMDA, and DA receptors expression in GLP-1-treated SH-SY5Y cells. Immunoblotting results indicated that vimentin (Vim) levels were downregulated in response to shorter GLP-1 treatment durations (**a**). In contrast, the α-amino-3-hydroxy-5-methyl-4-isoxazolepropionic acid receptor (AMPAR, **b**), N-methyl-D-aspartate receptor (NMDAR, **c**), and dopamine receptor D1 (D1R, **d**) were clearly upregulated. Interestingly, both muscarinic acetylcholine receptor (mAChR) and nicotinic acetylcholine receptor (nAChR) were also downregulated during GLP-1 induced differentiation (**e**,**f**). These results suggested that GLP-1 triggered neuronal differentiation towards glutamatergic and dopaminergic neurons, but not acetylcholinergic neurons. (M ± SD; * *p* < 0.05; ** *p* < 0.01; *** *p* < 0.001; compared to the value of control group, *n* = 6).

**Figure 4 biology-09-00348-f004:**
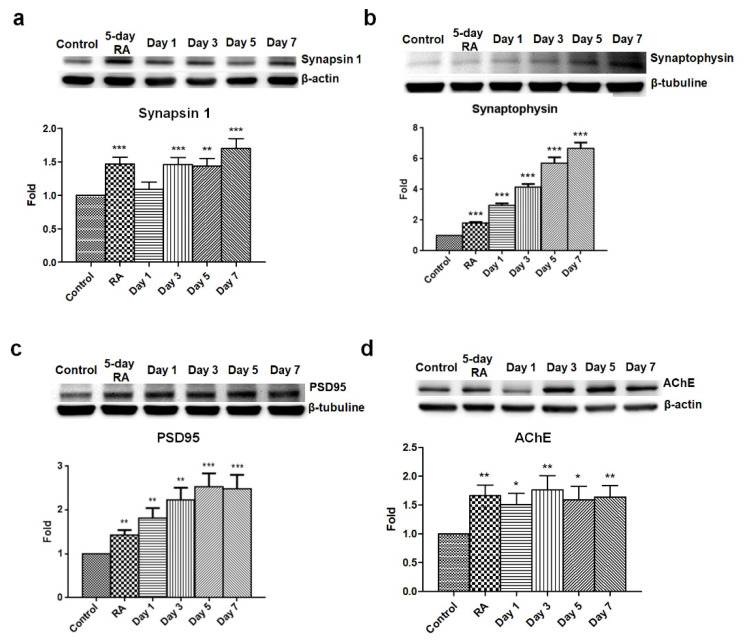
Elevated levels of synapsin 1, synaptophysin, PSD95, and AChE in GLP-1 differentiated SH-SY5Y neurons. Immunoblotting results detecting the expression levels of synapsin 1 (**a**), synaptophysin (**b**), PSD95 postsynaptic density protein 95, (**c**), and AChE acetylcholine esterase (**d**) revealed that these were significantly elevated in GLP-1 differentiated SH-SY5Y cells. These results suggested that GLP-1 administration resulted in enhanced synaptic development. (M ± SE; * *p* < 0.05; ** *p* < 0.01; *** *p* < 0.001; compared to the value of control group, *n* = 6).

**Figure 5 biology-09-00348-f005:**
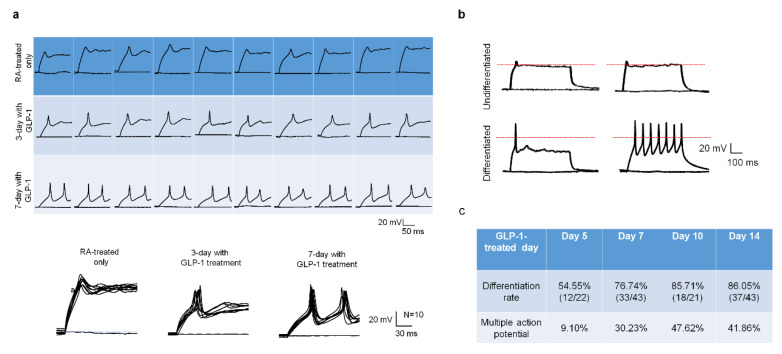
GLP-1 differentiated SH-SY5Y neurons have neuronal action potentials in response to electrical stimulation. The single cell patch clamp technique was employed to determine the electrophysiological function of GLP-1 differentiated SH-SY5Y neurons. The retinoic acid (RA, non-GLP-1) treated (undifferentiated) cells showed no response to stimulation, and 3-day GLP-1 treated SH-SY5Y cells (intermediate differentiated) started showing responses until 7-day GLP-1 treated (differentiated) SH-SY5Y cells had immediate action potential responses to stimuli (**a**). The neuronal action potential was also used to determine the maturity of GLP-1 differentiated neurons. The differentiated mature (7-day GLP-1) neurons showed responses to single and multiple stimuli, but undifferentiated (5-day RA) cells did not (**b**). The longer GLP-1 treatment resulted in higher percentage of mature neurons (**c**). These results suggested that GLP-1 administration was able to induce physiologically functional neurons. (the red dotted line shows V = 0).

**Figure 6 biology-09-00348-f006:**
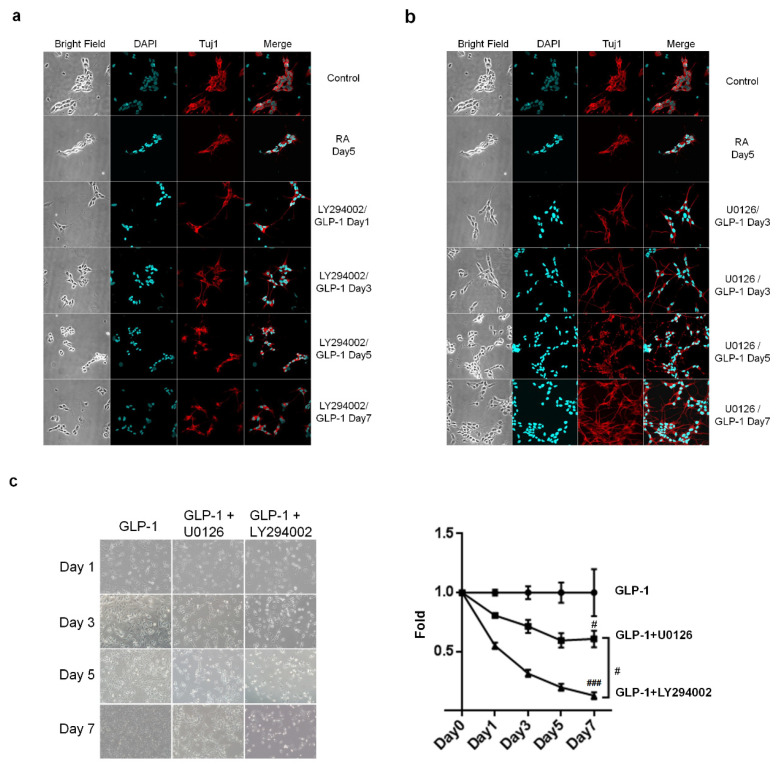
The PI3K-AKT signaling axis downstream of GLP-1R in the regulation of SH-SY5Y neuronal differentiation. The inhibitors of PI3K and MEK, LY294002 and U0126, respectively, were added to the medium, together with GLP-1, to assess the role of the GLP-1 signaling axis in the regulation of neuronal differentiation. Immunostaining images showed that LY294002 (**a**) inhibitor effectively suppressed neuronal differentiation and led to significant cell death (**b**). The U0126 inhibitor did not affect neuronal differentiation (**a**), although significant cell death was observed (**b**). The cell viability assay also has shown that LY294002 induced more cell death than U0126 (**c**). The results suggested that the PI3K-AKT-CREB axis is the predominant downstream signaling pathway regulating SH-SY5Y neuronal differentiation. (M ± SE; # *p* < 0.05; ### *p* < 0.001; compared to the value of GLP-1 control group, *n* = 3).

**Figure 7 biology-09-00348-f007:**
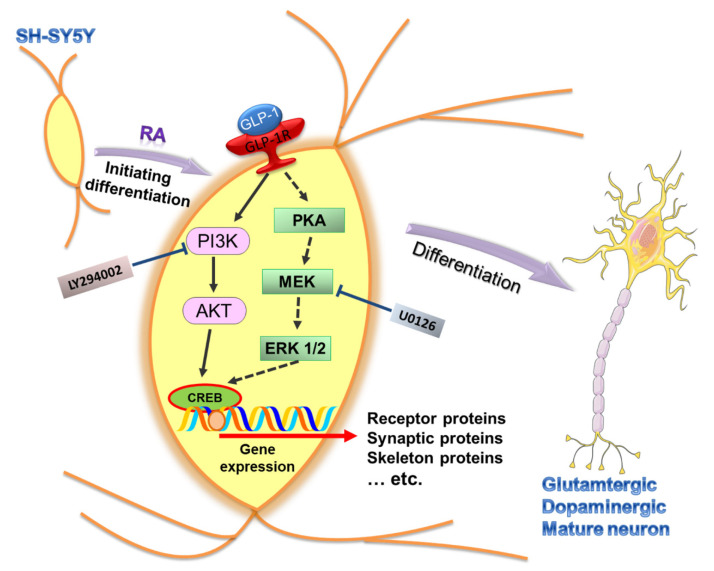
Schematic diagram illustrating the downstream signaling of activated GLP-1R for regulating the SH-SY5Y neuronal differentiation. The expression of GLP-1 receptors (GLP-1Rs) has been seen in human SH-SY5Y neuroblastoma cells and is involved in neuronal differentiation. Stimulation of GLP-1Rs triggers two downstream signaling axes, including PI3K-AKT (filled-line arrow) and PKA-ERK (dotted-line arrow). The specific inhibitors, LY294002 and U0126, were used to suppress the activity of PI3K and MEK, respectively. Our study’s overall results suggested that PI3K-AKT is the dominant downstream signaling axis regulating SH-SY5Y differentiation. In addition, the cells tend to develop into glutamatergic and dopaminergic neurons in GLP-1 induced neuronal differentiation. (RA: retinoic acid; cAMP: cyclic adenosine monophosphate; PKA: protein kinase A; MEK: mitogen-activated protein kinase kinase; ERK 1/2: mitogen-activated protein kinase 1 and 2; PI3K: phosphatidylinositide 3-kinase; AKT: protein kinase B; CREB: cAMP response element binding protein).

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
