# Peer review of "The Neurotrophic Function of Glucagon-Like Peptide-1 Promotes Human Neuroblastoma Differentiation via the PI3K-AKT Axis"

_biology, 2020, doi:10.3390/biology9110348_

Round 1

Reviewer 1 Report

In the paper "The Neurotrophic Function of Glucagon-like 2 Peptide-1 Promotes Human Neuroblastoma 3 Differentiation via the PI3K-AKT Axis" by Yang et al. the authors investigate how the GLP-1 receptor might impact neuronal health. The study is interesting and timely given the recent growing focus of GLP-1 receptor agonists in neurological diseases. The paper is extremely well written and is publishable with minor revisions.

Minor revision:
-Quantification of TUJ1 staining in Figure 2 would be beneficial.
-Since serum starvation inhibits mTOR and PI3K stimulates mTOR it would be interesting to look at mTOR in your setup. Perhaps you need PI3K stimulation but reduced mTOR (as happens with serum starvation) to induce differentiation. Does rapamycin treatment inhibit or stimulate differentiation? If possible, a western blot showing mTOR changes in your setup (+/- inhibitors etc) would be sufficient. 

Author Response

In the paper "The Neurotrophic Function of Glucagon-like 2 Peptide-1 Promotes Human Neuroblastoma 3 Differentiation via the PI3K-AKT Axis" by Yang et al. the authors investigate how the GLP-1 receptor might impact neuronal health. The study is interesting and timely given the recent growing focus of GLP-1 receptor agonists in neurological diseases. The paper is extremely well written and is publishable with minor revisions.

Minor revision:
-Quantification of TUJ1 staining in Figure 2 would be beneficial.

Answer:

  1. Thank reviewer give us a chance to explain the reason that we did not put the quantitative results of TUJ1. We have tried to quantitate TUJ1 positive cells by flow cytometer (Gallios Flow Cytometer, Beckman Coulter), unfortunately, only the 5-day retinoic acid (RA) treated cells were around 90% or more. The results of flow cytometry showed that all groups of RA and GLP-1 treated cells obtained similar percentage of TUJ1 positive cells (around 88%-95%). When we were strictly gating, only the percentage of TUJ1 positive decreased, the results of all groups were still similar. So the flow cytometry is not a good methodology to quantitate TUJ1-stained maturity of SH-SY5Y neurons. We also measured the immunofluorescent images by high-content microscope (ImageXpress XLS, Molecular Devices) to count TUJ1 stained cell number, the high-content microscopy results were similar to the flow cytometry. Therefore, we decided only showing the immunofluorescent images for readers, that should be more clear.

-Since serum starvation inhibits mTOR and PI3K stimulates mTOR it would be interesting to look at mTOR in your setup. Perhaps you need PI3K stimulation but reduced mTOR (as happens with serum starvation) to induce differentiation. Does rapamycin treatment inhibit or stimulate differentiation? If possible, a western blot showing mTOR changes in your setup (+/- inhibitors etc) would be sufficient. 

Answer:

  1. We deeply appreciate giving us a wonderful advice. We found SH-SY5Y cells treated fetal bovine serum (FBS)-containing medium with retinoic acid (RA) were starting to change morphology and but kept proliferating capability. Interestingly, the FBS-containing medium with RA was replaced by serum-free Neurobasal medium with B27, the cells kept differentiation and started dying from 5th The additional GLP-1 in serum-free Neurobasal medium with B27 could not only prevent SH-SY5Y death and also prohibit cell proliferation. So, we are currently looking the possible mechanisms that involve in regulating proliferation and differentiation. The reviewer mentioned mTOR may have important roles and we definitely will check mTOR signaling axis. We are thankful for reviewer’s good suggestion. We hope will finish this study and submit the manuscript in the near future.

Reviewer 2 Report

In my opinion, the article entitled The Neurotrophic Function of Glucagon-like Peptide-1 Promotes Human Neuroblastoma Differentiation via the PI3K-AKT Axis, is a very good work and it is suitable for publication in Biology with major revision. The results are very well showed, documented and the text is properly written. Figures are clear although there is a mistake in relation to Figures 5 and 6. Moreover, protocols of differentiation should be better described.

Major points:

Although data presented show that GLP-1 promotes differentiation efficiently, in my opinion, differentiated cells must be quantified and length of the neurites must be measured.

Minor points:

Material and Methods

  1. In general, I do not understand very well how exactly you get differentiation. In line 216 of Results you only specify serum-free Neurobasal with B27:
  2. Line 135: What is Neurobasal medium?
  3. Line 216: What is B27?

Results

  1. Line 211-212 and Fig. 1a: You say that TUJ1 is slightly expressed, but picture does not reflect this affirmation. GLPR1 is evident, but not TUJ1. You must include another picture.
  2. Figure 2: which is the difference between Control and 5day-RA cells? Both are apparently identical.
  3. Line 273-274: You affirm that The vanished expression of vimentin means that GLP-1 treated SH-SY5Y cells had a less mitotic and 273 more differentiated/mature neuron-like status. Have you performed any assay with neurofilament to confirm differentiation?
  4. Figure 4: 5 day-RA has the same effect over Synapsin 1 and AChE expression, which is the role of GLP-1 in these cases?
  5. Line 346-356: I think that when you say Fig. 6 in the text you mean Fig. 5.

References

I think that bibliography should be improved with most recent articles. For example, the review herewith incorporated is from 2019 and it is not included

This reference could be important:

Mol Metab. 2019 Dec; 30: 72–130.

Published online 2019 Sep 30. doi: 10.1016/j.molmet.2019.09.010

PMCID: PMC6812410

PMID: 31767182

Glucagon-like peptide 1 (GLP-1)

T.D. Müller,1,2,3,∗ B. Finan,4 S.R. Bloom,5 D. D'Alessio,6 D.J. Drucker,7 P.R. Flatt,8 A. Fritsche,2,9,10 F. Gribble,11 H.J. Grill,12 J.F. Habener,13 J.J. Holst,14 W. Langhans,15 J.J. Meier,16 M.A. Nauck,17 D. Perez-Tilve,18 A. Pocai,19 F. Reimann,11 D.A. Sandoval,20 T.W. Schwartz,21,22 R.J. Seeley,20 K. Stemmer,1,2 M. Tang-Christensen,23 S.C. Woods,24 R.D. DiMarchi,4,25 and M.H. Tschöp2,26,2

Author Response

In my opinion, the article entitled The Neurotrophic Function of Glucagon-like Peptide-1 Promotes Human Neuroblastoma Differentiation via the PI3K-AKT Axis, is a very good work and it is suitable for publication in Biology with major revision. The results are very well showed, documented and the text is properly written. Figures are clear although there is a mistake in relation to Figures 5 and 6. Moreover, protocols of differentiation should be better described.

Major points:

Although data presented show that GLP-1 promotes differentiation efficiently, in my opinion, differentiated cells must be quantified and length of the neurites must be measured.

Answer:

  1. Thank reviewer for mentioning the point and give us a chance to explain the reason that we did not put the quantitative results of TUJ1. One of other reviewers also pointed out the same question. We have tried to quantitate TUJ1 positive cells by flow cytometer (Gallios Flow Cytometer, Beckman Coulter), unfortunately, only the 5-day retinoic acid (RA) treated cells were around 90% or more. The results of flow cytometry showed that all groups of RA and GLP-1 treated cells obtained similar percentage of TUJ1 positive cells (around 88%-95%). When we were strictly gating, only the percentage of TUJ1 positive decreased, the results of all groups were still similar. So the flow cytometry is not a good methodology to quantitate TUJ1-stained maturity of SH-SY5Y neurons. We also measured the immunofluorescent images by high-content microscope (ImageXpress XLS, Molecular Devices) to count TUJ1 stained cell number, the high-content microscopy results were similar to the flow cytometry. We also used ImageJ to measure the length of neurites, but there were too much neurites overlap to measure, especially the 5-day and 7-day GLP-1 treated groups. Therefore, we decided only showing the immunofluorescent images for readers to judge, that should be more clear.

Minor points:

Material and Methods

  1. In general, I do not understand very well how exactly you get differentiation. In line 216 of Results you only specify serum-free Neurobasal with B27:

Answer:

The sentence of line 216 should be read starting line 214-216 described “Furthermore, to test whether the GLP-1 better induces the differentiation of SH-SY5Y cells than a previous study [34], we carried out an additional 5-day retinoic acid (RA) precondition and replaced the culture medium to serum-free Neurobasal with B27 supplement.”  The study of Luciani (ref 34) differentiates SH-SY5Y cells in FBS-containing medium with GLP-1 analogue (exendin-4), but we replaced “FBS-containing medium” with “serum-free Neurobasal medium with B27 supplement”.

  1. Line 135: What is Neurobasal medium?

Answer:

Neurobasal Medium is a basal medium produced by Gibco/Thermo Fisher, which is designed for long-term maintenance and maturation of pure pre-natal and embryonic neuronal cell populations without the need for an astrocyte feeder layer when used with Gibco/Thermo Fisher B-27 supplements. The components of medium include amino acids, vitamins, inorganic salts, glucose, sodium pyruvate, HEPES, and phenol red. The details of components are revealed in the website link https://www.thermofisher.com/tw/zt/home/technical-resources/media-formulation.251.html.

  1. Line 216: What is B27?

Answer:

B-27 supplement is also a product of Gibco/Thermo Fisher, which is an optimized serum-free supplement used to support the culture of embryonic, post-natal, and adult hippocampal, and other CNS neurons. The ingredient of B27 is proprietary of the manufactory. B27 contents 21 components including BSA, fatty acids, vitamins, anti-oxidants…etc, but the detail of contents are not fully revealed. 

Results

  1. Line 211-212 and Fig. 1a: You say that TUJ1 is slightly expressed, but picture does not reflect this affirmation. GLPR1 is evident, but not TUJ1. You must include another picture.

Answer:

The original image is 1600X1600 pixel, but when the image was shrunk to published size that very little TUJ1 staining was gone. We attached the originally enlarged image as attached file.

  1. Figure 2: which is the difference between Control and 5day-RA cells? Both are apparently identical.

Answer:

Thanks for reviewer’s suggestion. Since some longer cultured control cells also grew little neuronal process, to avoid the confuse, we replaced the images of control group to better representative images. (see attached file)

  1. Line 273-274: You affirm that The vanished expression of vimentin means that GLP-1 treated SH-SY5Y cells had a less mitotic and 273 more differentiated/mature neuron-like status. Have you performed any assay with neurofilament to confirm differentiation?

Answer:

As mentioned above, we have tried using ImageJ to measure the length of neurites but it was not that successful. So we the quantitated the synaptic proteins and examined stimulating action potential to reinforce the immunofluorescent images.

  1. Figure 4: 5 day-RA has the same effect over Synapsin 1 and AChE expression, which is the role of GLP-1 in these cases?

Answer:

According to the Western blotting results, which suggests that retinoic acid (RA) has major role on initially upregulating synapsin 1 and AchE, but when RA was replaced by GLP-1 the protein levels were maintained similar (AChE) or slightly increased (synapsin 1). So, GLP-1 is more likely maintaining than promoting expression on synapsin 1 and AChE.    

  1. Line 346-356: I think that when you say Fig. 6 in the text you mean Fig. 5.

Answer:

Thanks to reviewer carefully read and pointed out the mistake of misplace the figure number. The mistake was corrected.

References

I think that bibliography should be improved with most recent articles. For example, the review herewith incorporated is from 2019 and it is not included

This reference could be important:

Mol Metab. 2019 Dec; 30: 72–130.

Published online 2019 Sep 30. doi: 10.1016/j.molmet.2019.09.010

PMCID: PMC6812410

PMID: 31767182

Glucagon-like peptide 1 (GLP-1)

T.D. Müller,1,2,3, B. Finan,4 S.R. Bloom,5 D. D'Alessio,6 D.J. Drucker,7 P.R. Flatt,8 A. Fritsche,2,9,10 F. Gribble,11 H.J. Grill,12 J.F. Habener,13 J.J. Holst,14 W. Langhans,15 J.J. Meier,16 M.A. Nauck,17 D. Perez-Tilve,18 A. Pocai,19 F. Reimann,11 D.A. Sandoval,20 T.W. Schwartz,21,22 R.J. Seeley,20 K. Stemmer,1,2 M. Tang-Christensen,23 S.C. Woods,24 R.D. DiMarchi,4,25 and M.H. Tschöp2,26,2

Answer:

We’d like thank to reviewer’s suggestion. The review article gives thoroughly information of GLP-1We have incorporated the article as a reference in the end of second paragraph in “Introduction section” (highlighted).

Reviewer 3 Report

Good, interesting paper- congratulations. 

I have only one remark- The Authors did 6 Immunoblots to detect the expression levels of synapsin 1 (panel a), synaptophysin (panel b), PSD95 (postsynaptic density protein 95, panel c), and AChE (acetylcholine esterase, panel d). I'm not convinced by the results for synaptophysin (panel b) and PSD95 (postsynaptic density protein 95, panel c). The background is huge, especially for last three rows. If these are best representative images from six performed I do not believe in significances and data obtained for these two proteins. Therefore I'm also not convinced with the statement that 'GLP-1 Upregulates Synaptic Proteins". 

In this case it would be advisable to describe more the Immunoblotting data calculations. 'Immunolabeled proteins were visualized by 161 using an enhanced chemiluminescence kit (Amersham). Images were taken and analyzed by the UVP 162 ChemStudio PLUS Touch system (Analytik Jena)." Such description is not enough when such thick bands are presented for beta-tubilin serving as control and big background for analysed proteins.

Author Response

I have only one remark- The Authors did 6 Immunoblots to detect the expression levels of synapsin 1 (panel a), synaptophysin (panel b), PSD95 (postsynaptic density protein 95, panel c), and AChE (acetylcholine esterase, panel d). I'm not convinced by the results for synaptophysin (panel b) and PSD95 (postsynaptic density protein 95, panel c). The background is huge, especially for last three rows. If these are best representative images from six performed I do not believe in significances and data obtained for these two proteins. Therefore I'm also not convinced with the statement that 'GLP-1 Upregulates Synaptic Proteins". 

In this case it would be advisable to describe more the Immunoblotting data calculations. 'Immunolabeled proteins were visualized by 161 using an enhanced chemiluminescence kit (Amersham). Images were taken and analyzed by the UVP 162 ChemStudio PLUS Touch system (Analytik Jena)."  

Answer:

We are grateful that reviewer mentioned this point, so we have a chance to explain the difficulties. There are possibly two reasons that cause the strong background of synaptophysin and PSD95 in Western blots, low expression of proteins or low titer/low specificity of primary antibodies. We have tested 3 synaptophysin antibodies and 2 PSD95 antibodies, but still got faint and multiple non-specific bands. So, we only could choose a bad other than worse. Then, we increased the amount of total protein from 20 ug to 50 ug, but still took long exposure time to obtain better bands. That was the reason that we got strong background and thick actin bands. The differentiated SH-SY5Y neurons may not express much synaptophysin and PSD95 comparing with other synaptic proteins, synapsin 1 and AChE.

The UVP 162 ChemStudio PLUS Touch system is a CCCD image system taking digital images, the only thing we can do is to set the exposure time. The blot analyzing software of “UVP 162 ChemStudio PLUS Touch system” is automatically to subtract the background from selected bands and to analyze the intensity of bands, we could not do any image manipulation. The results are definitely trustable. We obtained 6 sets of images, we picked a set most fit the average trend, they may not perfect images but tell the truth. We think reviewer is a very experienced scientist should be very understanding of our point.  

Round 2

Reviewer 2 Report

You say:

We also used ImageJ to measure the length of neurites, but there were too much neurites overlap to measure, especially the 5-day and 7-day GLP-1 treated groups. Therefore, we decided only showing the immunofluorescent images for readers to judge, that should be more clear.

In my opinion, you should try to count differentiated cells and measure neurite lenght even if there is overlap. I think you must seed cells at lower density in order to visualize them better. It is a very important information to reinforce your biochemical results. As cell biology, it is very important quantify cell data. This is my suggestion.

Author Response

Reviewer 2

In my opinion, you should try to count differentiated cells and measure neurite length even if there is overlap. I think you must seed cells at lower density in order to visualize them better. It is a very important information to reinforce your biochemical results. As cell biology, it is very important quantify cell data. This is my suggestion.

Answer:

We appreciate reviewer’s persistent suggestion. We did our best to measure the neurite length of each group by ImageJ. We randomly picked 50 cells of each group and obtained the average neurite length. The result was made a table (as below) and added in the “Figure 2”, hope the quantitated results with the immunofluorescent images can give readers more clear understanding.

Control

RA

GLP1 Day1

GLP1 Day3

GLP1 Day5

GLP1 Day7

9.03±4.00 μm

10.01±4.29 μm

10.67±4.85* μm

11.96±6.87** μm

19.96±12.91*** μm

29.89±27.37*** μm

(the statistical significance was compared with control group, * 0.05>P, ** 0.01>P, *** 0.001>P)

Reviewer 3 Report

The explanation given satisfies me. The Authors did what could be done to get a good result. Therefore the paper is suitable for publication in its present form.

Author Response

We would like to thanks reviewer give us good advises.